# SARS-CoV-2 Neutralizing Antibodies to B.1 and to BA.5 Variant after Booster Dose of BNT162b2 Vaccine in HIV Patients COVID-Naïve and on Successful Antiretroviral Therapy

**DOI:** 10.3390/vaccines11040871

**Published:** 2023-04-20

**Authors:** Ilaria Vicenti, Monica Basso, Nicole Pirola, Beatrice Bragato, Maria Cristina Rossi, Mario Giobbia, Susanna Pascoli, Antonio Vinci, Sara Caputo, Ilenia Varasi, Camilla Biba, Lia Fiaschi, Maurizio Zazzi, Saverio Giuseppe Parisi

**Affiliations:** 1Department of Medical Biotechnologies, University of Siena, 53100 Siena, Italy; vicenti@unisi.it (I.V.); ilenia.varasi@gmail.com (I.V.); camilla.biba@student.unisi.it (C.B.); lia300790@gmail.com (L.F.); maurizio.zazzi@unisi.it (M.Z.); 2Department of Molecular Medicine, University of Padova, 35100 Padova, Italy; monica.basso@unipd.it (M.B.); nicole.pirola@studenti.unipd.it (N.P.); beatrice.bragato@studenti.unipd.it (B.B.); sara.caputo@studenti.unipd.it (S.C.); 3Infectious Diseases Unit, Treviso Hospital, 31100 Treviso, Italy; mariacristina.rossi@aulss2.veneto.it (M.C.R.); mario.giobbia@aulss2.veneto.it (M.G.); 4Microbiology Unit, Department of Specialist and Laboratory Medicine, Ca’ Foncello University Hospital, 31100 Treviso, Italy; susanna.pascoli@aulss2.veneto.it; 5Hospital Health Management Area, Local Health Authority “Roma 1”, Borgo Santo Spirito 3, 00193 Rome, Italy; antonio.vinci.at@hotmail.it

**Keywords:** BNT162b2 mRNA vaccine, third dose, HIV, SARS-CoV-2, B.1 lineage, BA.5 lineage, neutralizing antibodies

## Abstract

Live virus neutralization is the gold standard to investigate immunity. This prospective observational study aimed to determine the magnitude of response against the original B.1 lineage and against the BA.5 lineage six months after the third BNT162b2 mRNA vaccine dose in patients with HIV infection on successful antiretroviral treatment and no previous SARS-CoV-2 infection. A total of 100 subjects (M/F 83/17, median age 54 years) were included in the analysis: 95 had plasma HIV RNA <40 copies/mL, the median CD4+ T cell count at the administration of the third dose was 580 cells/mm^3^, and the median nadir CD4+ T cell count was 258 cells/mm^3^. Neutralizing antibodies (NtAb) against B.1 were detectable in all the subjects, but those to BA.5 were only detected in 88 (*p* < 0.001). The median NtAb titer to B.1 was significantly higher than that to BA.5 (393 vs. 60, *p* < 0.0001), and there was a strong positive correlation between the paired measurements (*p* < 0.0001). Linear regression on a subset of 87 patients excluding outlier NtAb titers showed that 48% of the changes in NtAb titers to BA.5 are related to the changes in value titers to B.1. SARS-CoV-2 variants evolve rapidly, challenging the efficacy of vaccines, and data on comparative NtAb responses may help in tailoring intervals between vaccine doses and in predicting vaccine efficacy.

## 1. Introduction

The continuous evolution of the SARS-CoV-2 virus has been a major challenge over the three years of the pandemic. The Omicron variant (B.1.1.529), first identified in South Africa in November 2021, rapidly replaced the previously dominant variants, becoming the fifth variant of concern (VOC) as declared by the WHO on 26 November 2021. Since its emergence, the Omicron variant (B.1.1.529) has evolved into a plethora of descendant sublineages, including BA.1, BA.2, BA.3, BA.4, and BA.5. The BA.5 sublineage has been dominant in Italy for several months, with an estimated prevalence higher than 90% up to October 2022 [1]. This subvariant is characterized by 30 mutations in the spike protein and about 20 in other sites of the genome with respect to the initial version of SARS-CoV-2 [2]. As a consequence, BA.5 can significantly escape neutralizing antibodies induced both by infection with previous VOCs and by vaccination, and a substantial proportion of the general population is not protected from BA.4/5-related hospitalization at less than 6 months after the third vaccine dose. Hachmann et al. [3] demonstrated that the neutralizing antibody titer against BA.5 was lower by a factor of 21 with respect to the titer against WA1/2020 and lower by a factor of 3.3 with respect to the titer against BA.1, and Tartof et al. [4] described adjusted effectiveness of 73% (95% CI 25–91) against hospitalisation and of 43% (95% CI 10–63) against emergency department admission. Two key mutations are involved in the immune escape shown by BA.5: F486V and L452R, present in the receptor binding domain (RBD) of the spike protein and able to reduce serum neutralization with respect to BA.1 and BA.2 in patients vaccinated with three doses. The first mutation, F486V, is an amino acid substitution not identified in other Omicron subvariants, and the second mutation, L452R, is different from L452Q identified in BA.2.12.1 because Q is an amino acid with no charge while L is an amino acid that has a hydrophobic side chain (probably more interferent with antibody binding) [5,6,7].

The CD4+ memory response to the BNT162b2 vaccine is a complex phenomenon: unlike the neutralizing response, the combination BNT162b2/BNT162b2 elicits a more efficient cellular response with respect to a combination adenovirus vector-based vaccine, AZD122/BNT162b2 [8]. A low decline of CD4+ T cell memory occurs throughout the months, but there is a total recovery after the third BNT162b2 vaccine dose [9]. Taken together, these data showed a successful memory response lasting many months in healthy subjects. In HIV-positive patients, the influence of viroimmunological parameters had to be taken into account: most subjects on successful ART with a high CD4+ cell count at inclusion and receiving two doses of the BNT162b2 vaccine had a cellular response after six months, and a high number of CD4+ cell count is a predicting factor [10]. A comparable T cell response after six months from the third dose of the BNT162b2 vaccine was described in patients with either less or more than 200 cells/mm^3^, but with virological suppression (median HIV RNA 40 copies/mL) [11]. These results suggest that a standard vaccine schedule could be applied to HIV patients with ongoing successful ART, but untreated or failed subjects may need a tailored vaccine administration.

Anti-SARS-CoV-2 vaccines are safe, and they were shown to reduce morbidity and mortality: real-word data confirmed the data reported in registration studies [12,13,14]. The third dose and its timing have a peculiar role: HIV-positive patients who received 2 vaccine doses are at higher risk of breakthrough infections with respect to HIV-negative subjects, but the third BNT162b2 dose has a protective role [15], and a higher neutralizing antibody seronversion rate was described in HIV-positive patients receiving the third dose of the inactivated COVID-19 vaccine after 5 months with respect to the 3-month interval, especially in subjects with a CD4 count <200 cells/mm^3^ [16].

Patients with HIV infection boosted with either heterologous or homologous COVID-19 vaccines showed a strong antibody response as evaluated with IgG ELISA against SARS CoV-2 spike (i.e., from a median of 2644 AU/mL 182 days after the first vaccine dose to a median of 143,088 AU/mL about a month post-third dose [17] and from a median of 107 AU/mL at pre-booster study time to a median of 1580 AU/mL four weeks after the third dose [18]), but a few data were reported on authentic virus neutralization response against the BA.5 variant after the third vaccine dose in patients receiving an mRNA vaccine. Previous studies described a neutralizing response following booster vaccination in subjects receiving an mRNA vaccine, but the targets were wild-type (Wuhan A), Delta (B.1.617.2; AY.2), and Omicron (B.1.1.529; BA.1) [19,20,21].

Therefore, the aim of this study was to determine the magnitude of SARS-CoV-2 neutralizing antibodies (NtAb) against the original B.1 lineage and against the BA.5 lineage six months after the third BNT162b2 mRNA vaccine dose in patients with HIV infection on antiretroviral treatment (ART).

## 2. Materials and Methods

### 2.1. Study Design and Population

The study was a prospective, observational, single cohort study approved by the “Comitato per la Sperimentazione Clinica di Treviso e Belluno” (protocol code 812/2020), and written informed consent was obtained from each participant.

The patient cohort included 100 subjects with HIV infection, aged more than 18 years, who completed a three-dose regimen of the BNT162b2 vaccine. Previous or present SARS-CoV-2 infection was excluded based on the clinical history and the lack of positive results on nasopharyngeal swabs analyzed for clinical needs (including contact tracing).

Patients were enrolled at the Infectious Diseases Unit of the Treviso Hospital, Treviso, Italy. There was no loss on follow-up.

The study was reported according to the Strengthening the Reporting of Observational Studies in Epidemiology (STROBE) Statement [22].

### 2.2. Variables and Assessment Methods

Plasma HIV viremia, CD4+ T cell count at the time of the third vaccine dose, CD4+ T cell count at nadir, and main clinical and demographic characteristics of the patients were collected. A successful HIV-1 suppression was defined when HIV RNA viremia was below 40 copies/mL.

Six months after the third BNT162b2 vaccine dose, NtAb titers were determined in a live virus microneutralization assay against the BA.5 omicron sublineage (EPI _ISL_14513768) (BA.5.1) and the B.1 wild-type SARS-CoV-2 strain (EPI_ISL_2472896) as previously described [23,24]. Briefly, two-fold serial dilutions of sera were incubated with 100 TCID_50_ of the corresponding SARS-CoV-2 strain and then added to pre-seeded Vero E6 cells and incubated for 72 h at 37 °C with 5% CO_2_. The virus cytopathic effect was quantified by measuring cell viability with the Cell-titer Glo 2.0 system in a GloMax Discover luciferase plate reader (Promega, Madison, WI, USA). The NtAb titer (ID_50_) was defined as the reciprocal value of the sample dilution that showed 50% protection from the virus-induced cytopathic effect. Each serum was tested in duplicate, and each run included an uninfected control, an infected control, and the virus back-titration to confirm the virus inoculum. The initial validation of the assay was performed with the First WHO International Standard anti-SARS-CoV-2 immunoglobulin (Version 3.0, Dated 17 December 2020; code 20/268 NIBSC, Ridge, UK) [25]. Sera with ID_50_ below 10 were defined as negative and scored as 5 for statistical analysis.

### 2.3. Statistical Analysis

Statistical analysis was carried out using MedCalc^®^ Statistical Software version 20.116 (MedCalc Software Ltd., Ostend, Belgium) and Stata v. 17.0. (StataCorp. College Station, TX, USA). Data were summarized using absolute and relative frequencies for categorical variables and median and IQR for numerical variables. Mann–Whitney U test, chi-squared test, and Fisher’s exact test were employed as appropriate. A spearman rank correlation was applied for the comparison of NtAb titers to B.1 and BA.5 lineages and for comparison with other continuous variables. The statistical significance level was set at α = 0.05 for all analyses.

### 2.4. Bias and Sensitivity Analysis

The main source of bias lies in the inadvertent inclusion of patients who had contracted an asymptomatic, undiagnosed SARS-CoV-2 infection. This would translate into erroneously including subjects with hybrid immunity (infection plus vaccination), which has been shown to be of higher magnitude and broader compared with the response to vaccine alone [14].

To reduce this potential source of bias, a sensitivity analysis was performed by excluding patients with NtAb titers exceedingly high within the cohort. The Grubbs test was applied to identify such outliers of NtAb titers to either B.1 or BA.5, independently [26]. Pearson’s correlation and linear regression analyses were then used on this subset.

## 3. Results

A total of 100 subjects (Male/Female 83/17, median age 54 years, IQR 49–60 years) were included in the analysis. All of them were on ART, most (68%) on triple drug regimens, and with suppressed HIV viremia (95%). The median CD4+ T cell count at the administration of the third dose was 580 cells/mm^3^ (IQR 411–786 cells/mm^3^), and the median nadir CD4+ T cell count was 258 cells/mm^3^ (IQR 117–360 cells/mm^3^). The characteristics of the study population are reported in Table 1.

The main results of our study were the stronger neutralizing response to B.1 with respect to BA.5, the positive correlation between the two titers, and the lack of correlation with CD4+ cell count.

NtAb against B.1 was detectable in all the subjects; conversely, 12 patients were negative for BA.5 (*p* < 0.001). Overall, the median NtAb titer to B.1 was significantly higher than that to BA.5 (393, IQR 128–866 vs. 60, IQR 28–143, *p* < 0.0001); however, there was a strong positive correlation between the paired measurements (r= 0.854, 95% CI 0.79–0.9, *p* < 0.0001) (Figure 1). The difference in NtAb titers between the two strains remained significant when subjects only scored positive for both lineages were analyzed (459 [IQR 162–907] to B.1 vs. 72 [36–204] to BA.5, *p* < 0.0001). Only one subject, an African male, showed NtAbs titers to BA.5 higher than to B.1 (2961 vs. 1378). In this case, the BA.5, but not the B.1, was recognized as an outlier.

Neither NtAb titers to B.1 nor to BA.5 showed any correlation with current CD4+ T cell counts (r = 0.0752, *p* = 0.5680; and r = −0.00461, *p* = 0.9721; respectively) or with CD4+ T cell counts at nadir (r = −0.0380, *p* = 0.7148; and r = −0.0426, *p* = 0.6819; respectively). NtAb titers to BA.5, but not to B.1, had a positive correlation with age at diagnosis (r = 0.276, 95% CI 0.08–0.450, *p* = 0.0063) and at enrollment (r = 0.227, 95% CI 0.0309–0.406, *p* = 0.0239).

### Sensitivity Analysis

Thirteen patients were excluded in a sensitivity analysis excluding outlier NtAb titers (2 to B.1, 5 to BA.5, and 6 to both strains; see Appendix A). The correlation between NtAb titers to B.1 and BA.5 was confirmed in the resulting subset of 87 patients (r = 0.69, *p* < 0.0001). The significantly higher NtAb titer to B.1 was confirmed in this subgroup of 87 patients (268, IQR 114–593 vs. 50, IQR 25–105, *p* < 0.0001). Linear regression showed that 48% of the changes in NtAb titers to BA.5 are related to the changes in value titers to B.1 (Figure 2).

A detailed comparison between the main results and the sensitivity analysis was reported in Appendix A.

## 4. Discussion

Here we describe the correlation between NtAb titers against B.1 and against BA.5 six months after the third dose of the BNT162b2 vaccine in a cohort of 100 HIV-positive subjects on successful ART and COVID-naïve.

NtAb titers to BA.5 were significantly lower with respect to titers to B.1 six months after the third dose of the BNT162b2 vaccine in a cohort of HIV-positive subjects, mostly on successful ART. This result is in agreement with previously reported data in patients receiving three vaccine doses and having NtAb tested about one month [5,27] and eight months [28] after the third dose. The subjects enrolled in the first study [27] were HIV-positive patients on ART with a median CD4+ T cell count > 500 cells/mm^3^, while subjects included in the studies by Cheng et al. [5] and by Anichini et al. [28] were likely HIV-uninfected (no information was reported). Taken together, all these results suggest that the striking divergence of Omicron BA.5 is the major driver for vaccine escape and that the NtAb response in immune restored HIV-positive subjects is similar to that occurring in HIV-negative patients [27,29]. The independence of the neutralizing activity to BA.5 from immunocompetence is in accord with the data reported in the study by Corma-Gomez et al. [30], who described a lower neutralizing activity to BA.5 with respect to B.1 from 4 to 8 weeks after the booster vaccine dose both in patients with CD4 counts < 200 cells/mm^3^ and in those with CD4 counts ≥ 200 cells/mm^3^.

Despite a significantly lower response to BA.5, there was a clear correlation between NtAb titers to B.1 and to BA.5, both in the overall study population and in the subset excluding subjects with outliers to either viral strain. Despite an overall low response, NtAb titers to BA.5 varied over three orders of magnitude. This might reflect the inclusion of subjects with an undiagnosed prior SARS-CoV-2 infection and/or higher cross-reactivity due to prior exposure to seasonal human coronaviruses [31]. The latter could explain the positive correlation between BA.5 NtAb titers and patient age detected in our case file, but this is a hypothesis that needs to be confirmed on a larger number of patients.

There is an increasing interest in the clinical role of a model able to predict the humoral response to SARS-CoV-2 vaccine in specific populations: Voutouri et al. [32] elaborated a mechanistic model to simulate the efficacy of a vaccine booster in immunosuppressed patients, and Parisi et al. [24] described the kinetics of the response to the ancestral virus in a cohort of healthcare workers infected during the first SARS-CoV-2 wave and monitored to the completion of their triple vaccination cycle.

Real-life investigations focused on the disease severity of the SARS-CoV-2 infection caused by the BA.5 strain sublineage reported quite different results. During the BA.1 and BA.4/BA.5 waves, the risk of admission to an intensive care unit, mechanical ventilation, or steroid prescription/death within 21 days from diagnosis was lower with respect to the ancestral, Beta, and Delta waves. In addition, vaccination had a protective role [33], and a considerable risk reduction of death outcome associated with a booster vaccine dose was observed either for BA.5 or BA.2 related infections [34]. Conversely, a Danish nationwide population-based study described a higher hospitalization rate in subjects with BA.5 infection with respect to subjects with BA.2 infection (1.9% vs. 1.4%) [35]: BA.5 infection was also associated with an 18% increased rate of hospitalization and a longer duration of symptoms compared to BA.1 [36,37].

The main limitations of the study are the lack of patients with a previous SARS-CoV-2 infection, the lack of patients vaccinated with the new bivalent mRNA formulations expressing BA.5, and the lack of a control group. Furthermore, it has to be underlined the unavailability of a longitudinal evaluation of neutralizing response and cellular analysis of T-cell memory response: we did not include these data in our study because of the difficulty of obtaining a large amount of blood from patients who are already regularly monitored and because of the elevated costs of such studies, which make it necessary to work in a biosafety level 3 laboratory. We identified COVID-19-naïve subjects based on self-reported data, as currently accepted [18,38]: antibody titers against the N antigen protein waned with time [39,40], and they could not be detectable after SARS-CoV-2 infection [41,42]. 

On the other hand, this study included 100 HIV-positive subjects, all vaccinated with the same mRNA vaccine and regularly followed, who are an ideal cohort for prolonged monitoring of humoral responses and for assessing an overtime titer threshold for immunological protection from BA.5 infection [43]. Of note, in our study, the interval between booster vaccine dose and neutralizing activity testing is six months, so that giving updated information on the humoral response to the wild-type virus and to BA.5 is at a time corresponding to the administration of the fourth vaccine dose [44]. Future evaluations of the cohort enrolled in this study will be aimed at studying neutralizing response durability in persistently COVID-naïve patients and modification over time of the titer in breakthrough cases. The availability of studies on the humoral response in HIV patients receiving the same vaccine schedule but with different viroimmunological characteristics (i.e., failed subjects) or with comorbidities modifying the immune response (i.e., cancer on chemotherapy) could help to individualize the risk of severe disease in these frail subjects and to prescribe the more appropriate treatment in the case of SARS-CoV-2 infection.

## 5. Conclusions

SARS-CoV-2 variants evolve rapidly, challenging the efficacy of vaccines and data on comparative NtAb response may help in tailoring intervals between vaccine doses and in predicting vaccine efficacy. NtAb titers are the best correlate of protection from severe COVID-19 [45] and live virus neutralization, despite the time consuming and requiring biosafety level 3 containment, remains the gold standard to investigate immunity in specific cohorts. In the HIV-positive population, further studies are needed to define the role of the fourth vaccine dose and what immunovirological parameters may impact the success of vaccination [38].

## Figures and Tables

**Figure 1 vaccines-11-00871-f001:**
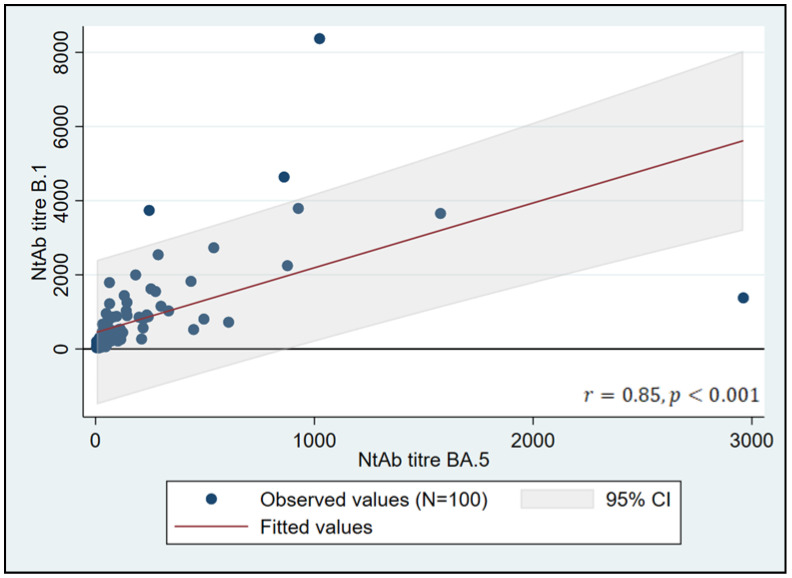
Linear regression between B.1 titers and BA.5 titers in HIV patients 6 months after the third BNT162b2 vaccine dose (entire cohort). NtAb titers are expressed as ID_50_, which is the reciprocal value of the sera dilution and shows 50% protection from the virus-induced cytopathic effect.

**Figure 2 vaccines-11-00871-f002:**
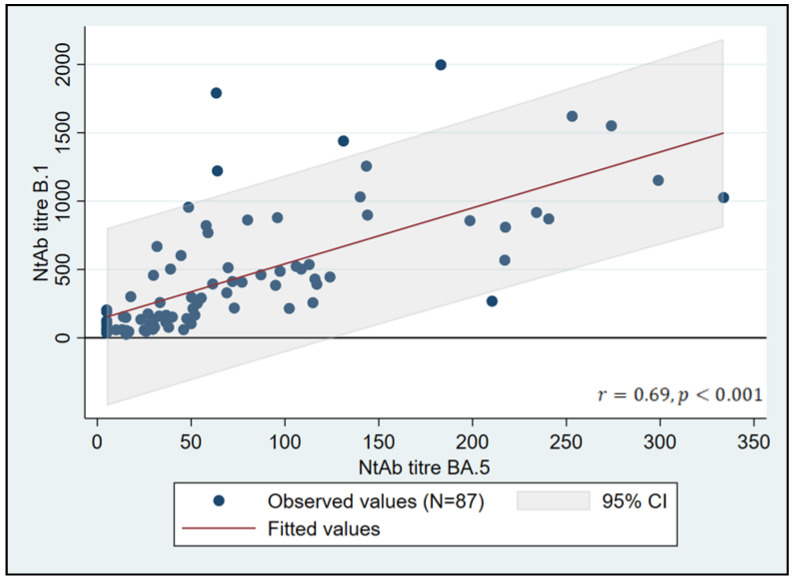
Linear regression between B.1 and BA.5 titers in HIV patients 6 months after the third BNT162b2 vaccine dose (outliers excluded from cohort). NtAb titers are expressed as ID_50_, which is the reciprocal value of the sera dilution and shows 50% protection from the virus-induced cytopathic effect.

**Table 1 vaccines-11-00871-t001:** Main characteristics of the study population (*n* = 100).

Male, *n* (%)	83 (83)
Caucasians, *n* (%)	88 (88)
Age at HIV diagnosis (years), median (IQR)	36 (28–48)
Age at study enrollment (years), median (IQR)	54 (49–60)
Absolute number of CD4+ T cell count at nadir (cells/mm^3^), median (IQR)	258 (117–360)
Patients with <200 CD4+ T cell count at nadir, *n* (%)	36 (36)
Current CD4+ T cell count (cells/mm^3^), median (IQR)	580 (411–786)
Patients with suppressed undetectable plasma HIV RNA, *n* (%)	95 (95)
HIV RNA value in patients with detectable HIV viremia (copies/mL), median (IQR)	58 (49–72)

## Data Availability

The raw data on demographics and clinical status of participants are protected and not available due to data privacy laws. The processed data are available from the corresponding author upon reasonable request.

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
