# Peer review of "SARS-CoV-2 Neutralizing Antibodies to B.1 and to BA.5 Variant after Booster Dose of BNT162b2 Vaccine in HIV Patients COVID-Naïve and on Successful Antiretroviral Therapy"

_vaccines, 2023, doi:10.3390/vaccines11040871_

Round 1
Reviewer 1 Report
The manuscript represents an exciting analysis of the response to the COVID-19 vaccine and HIV infection. The manuscript, however only describes the presence of neutralizing antibodies against SARS CoV-2 in stable HIV patients. It would be interesting to have a longitudinal report on the presence of neutralizing antibodies and a cellular analysis of T-cell memory response. The authors should discuss how lasting the CD4 memory response would be and what vaccine schemes should be employed. Finally, the introduction should mention several articles on vaccination in HIV individuals for exampledoi: 10.3389/fimmu.2023.1136723. eCollection 2023. doi: 10.1097/COH.0000000000000790 doi: 10.3389/fimmu.2023.1152695. eCollection 2023.
Reviewer 2 Report
1. Please make the title short and comprehensive2. Please mention what BA stands for
3. Please elaborate your results about assessment here simply, before going to stat results
4. Please add short introductory para before details discussion
5. Please add some future recommendation at end of discussion section

Reviewer 3 Report
This is an important, well planned, conducted and reported study.
Introduction:
You wrote: “As a consequence, BA.5 can signifi- 45 cantly escape neutralizing antibodies induced both by infection with previous VOCs and 46 by vaccination and a substantial proportion of the general population is not protected 47 from BA.4/5-related hospitalization at less than 6 months after the third vaccine dose [3,4]. 48 Patients with HIV infection boosted with either heterologous or homologous 49 COVID-19 vaccines showed a strong antibody response as evaluated with IgG ELISA 50 against SARS CoV-2 spike [5,6] but a few data were reported on authentic virus neutrali- 51 zation response against BA.5 variant after the third vaccine dose in patients receiving a 52 mRNA vaccine. Previous studies described neutralizing response following booster vac- 53 cination in subjects receiving a mRNA vaccine but the targets were wild-type (Wuhan A), 54 Delta (B.1.617.2;AY.2), Omicron (B.1.1.529; BA.1) [7–9].”
[Can you please provide more numerical data from these studies as this is key to demonstrating how your study adds important data]
You wrote: “The BA.5 sublineage has been dominant in Italy for several months, 42 with an estimated prevalence higher than 90% up to October 2022 [1]. This subvariant is 43 charcterized by 30 mutations in the spike protein and about 20 in other sites of the genome. As a consequence, BA.5 can signifi- 45 cantly escape neutralizing antibodies induced both by infection with previous VOCs and 46 by vaccination and a substantial proportion of the general population is not protected 47 from BA.4/5-related hospitalization at less than 6 months after the third vaccine dose [3,4]”
[Do you have any data from other studies on which mutations are key to helping the BA.5 sublineage escape neutralizing antibodies?]
Methods
You wrote: “The main source of bias lies in the inadvertent inclusion of patients who had con- 103 tracted asymptomatic undiagnosed SARS-CoV-2 infection. This would translate into er- 104 roneously including subjects with hybrid immunity (infection plus vaccination) which has 105 been shown to be of higher magnitude and broader compared with response to vaccine 106 only [14]. 107 To reduce this potential source of bias, a sensitivity analysis was performed by ex- 108 cluding patients with NtAb titers exceedingly high within the cohort. Grubbs test was 109 applied to identify such outliers of NtAb titers to either B.1 or BA.5, independently [15]. 110 Pearson’s correlation and linear regression analysis were then used on this subset.”
[Good, this sensitivity analysis is an important maneovre]
Reviewer 4 Report
While this is a reasonably good clinical and experiemtntal study, I have a few concerns that need to be addressed. My main concern is that I don't think that the statistical tools used have been optimally or correctly utilized.
1) r (correlation coefficient) or r2 (coefficient coefficient) is nowhere seen in the manuscript. This is important in regressional analysis as it is an indicator of the strengh of the correlation, if any.
2) The authors mentioned "correlation" throughout the paper but technically and statistically, you cannot claim correlation inless you can show r >=0.5 or r2 >= 0.25. You can try to claim link or relationship with P < 0.05.
3) Both p values and r or r2 should appear on the figures.
4) r2 or r is important in this study since you can tell if the correlation is stronger in the HIV group or not. If weaker, you can argue that maybe the HIV group is not efficiently producing the necessary anitobies for some reasons. Maybe because of their impaired immune system? Or the antiretroviral drugs?
5) I believe that "Subvariants" instead of "Variant" is correct. We are referring to two Omicron subvariants. It should be plural as we are referring to two subvariants
Round 2
Reviewer 4 Report
Improvements seen